# Dynamic Network Surgery for Efficient DNNs

**Yiwen Guo**[*]
Intel Labs China
yiwen.guo@intel.com

**Anbang Yao**
Intel Labs China
anbang.yao@intel.com

**Yurong Chen**
Intel Labs China
yurong.chen@intel.com

## Abstract

Deep learning has become a ubiquitous technology to improve machine intelligence. However, most of the existing deep models are structurally very complex, making them difficult to be deployed on the mobile platforms with limited computational power. In this paper, we propose a novel network compression method called dynamic network surgery, which can remarkably reduce the network complexity by making on-the-fly connection pruning. Unlike the previous methods which accomplish this task in a greedy way, we properly incorporate connection splicing into the whole process to avoid incorrect pruning and make it as a continual network maintenance. The effectiveness of our method is proved with experiments. Without any accuracy loss, our method can efficiently compress the number of parameters in LeNet-5 and AlexNet by a factor of $108\times$ and $17.7\times$ respectively, proving that it outperforms the recent pruning method by considerable margins. Code and some models are available at https://github.com/yiwenguo/Dynamic-Network-Surgery.

## 1 Introduction

As a family of brain inspired models, deep neural networks (DNNs) have substantially advanced a variety of artificial intelligence tasks including image classification [13, 19, 11], natural language processing, speech recognition and face recognition.

Despite these tremendous successes, recently designed networks tend to have more stacked layers, and thus more learnable parameters. For instance, AlexNet [13] designed by Krizhevsky et al. has 61 million parameters to win the ILSVRC 2012 classification competition, which is over 100 times more than that of LeCun's conventional model [15] (e.g., LeNet-5), let alone the much more complex models like VGGNet [19]. Since more parameters means more storage requirement and more floating-point operations (FLOPs), it increases the difficulty of applying DNNs on mobile platforms with limited memory and processing units. Moreover, the battery capacity can be another bottleneck [9].

Although DNN models normally require a vast number of parameters to guarantee their superior performance, significant redundancies have been reported in their parameterizations [4]. Therefore, with a proper strategy, it is possible to compress these models without significantly losing their prediction accuracies. Among existing methods, network pruning appears to be an outstanding one due to its surprising ability of accuracy loss prevention. For instance, Han et al. [9] recently propose to make "lossless" DNN compression by deleting unimportant parameters and retraining the remaining ones (as illustrated in Figure 1(b)), somehow similar to a surgery process.

However, due to the complex interconnections among hidden neurons, parameter importance may change dramatically once the network surgery begins. This leads to two main issues in [9] (and some other classical methods [16, 10] as well). The first issue is the possibility of irretrievable network

---

[*]This work was done when Yiwen Guo was an intern at Intel Labs China supervised by Anbang Yao who is responsible for correspondence.

damage. Since the pruned connections have no chance to come back, incorrect pruning may cause severe accuracy loss. In consequence, the compression rate must be over suppressed to avoid such loss. Another issue is learning inefficiency. As in the paper [9], several iterations of alternate pruning and retraining are necessary to get a fair compression rate on AlexNet, while each retraining process consists of millions of iterations, which can be very time consuming.

In this paper, we attempt to address these issues and pursue the compression limit of the pruning method. To be more specific, we propose to sever redundant connections by means of continual network maintenance, which we call dynamic network surgery. The proposed method involves two key operations: **pruning** and **splicing**, conducted with two different purposes. Apparently, the pruning operation is made to compress network models, but over pruning or incorrect pruning should be responsible for the accuracy loss. In order to compensate the unexpected loss, we properly incorporate the splicing operation into network surgery, and thus enabling connection recovery once the pruned connections are found to be important any time. These two operations are integrated together by updating parameter importance whenever necessary, making our method dynamic.

In fact, the above strategies help to make the whole process flexible. They are beneficial not only to better approach the compression limit, but also to improve the learning efficiency, which will be validated in Section 4. In our method, pruning and splicing naturally constitute a circular procedure and dynamically divide the network connections into two categories, akin to the synthesis of excitatory and inhibitory neurotransmitter in human nervous systems [17].

The rest of this paper is structured as follows. In Section 2, we introduce the related methods of DNN compression by briefly discussing their merits and demerits. In Section 3, we highlight our intuition of dynamic network surgery and introduce its implementation details. Section 4 experimentally analyses our method and Section 5 draws the conclusions.

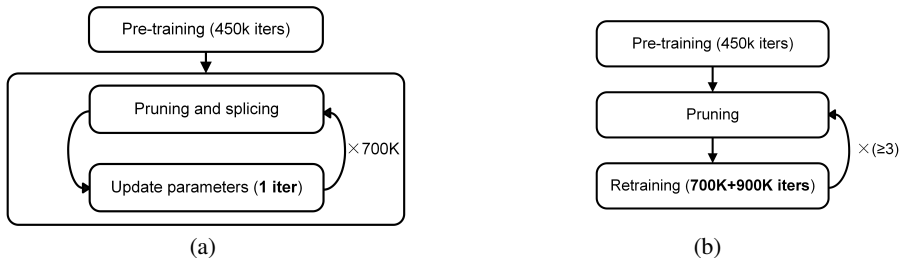

Figure 1: The pipeline of (a) our dynamic network surgery and (b) Han et al.'s method [9], using AlexNet as an example. [9] needs more than 4800K iterations to get a fair compression rate ($9\times$), while our method runs only 700K iterations to yield a significantly better result ($17.7\times$) with comparable prediction accuracy.

## 2  Related Works

In order to make DNN models portable, a variety of methods have been proposed. Vanhoucke et al. [20] analyse the effectiveness of data layout, batching and the usage of Intel fixed-point instructions, making a $3\times$ speedup on x86 CPUs. Mathieu et al. [18] explore the fast Fourier transforms (FFTs) on GPUs and improve the speed of CNNs by performing convolution calculations in the frequency domain.

An alternative category of methods resorts to matrix (or tensor) decomposition. Denil et al. [4] propose to approximate parameter matrices with appropriately constructed low-rank decompositions. Their method achieves $1.6\times$ speedup on the convolutional layer with 1% drop in prediction accuracy. Following similar ideas, some subsequent methods can provide more significant speedups [5, 22, 14]. Although matrix (or tensor) decomposition can be beneficial to DNN compression and speedup, these methods normally incur severe accuracy loss under high compression requirement.

Vector quantization is another way to compress DNNs. Gong et al. [6] explore several such methods and point out the effectiveness of product quantization. HashNet proposed by Chen et al. [1] handles network compression by grouping its parameters into hash buckets. It is trained with a standard backpropagation procedure and should be able to make substantial storage savings. The recently

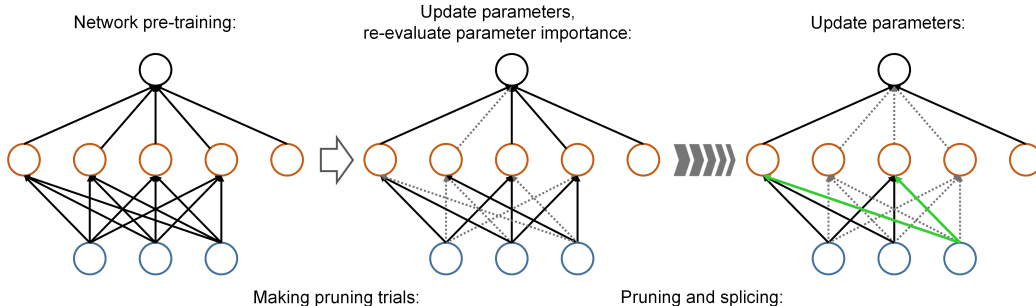

Figure 2: Overview of the dynamic network surgery for a model with parameter redundancy.

proposed BinaryConnect [2] and Binarized Neural Networks [3] are able to compress DNNs by a factor of $32\times$, while a noticeable accuracy loss is sort of inevitable.

This paper follows the idea of network pruning. It starts from the early work of LeCun et al.'s [16], which makes use of the second derivatives of loss function to balance training loss and model complexity. As an extension, Hassibi and Stork [10] propose to take non-diagonal elements of the Hessian matrix into consideration, producing compression results with less accuracy loss. In spite of their theoretical optimization, these two methods suffer from the high computational complexity when tackling large networks, regardless of the accuracy drop. Very recently, Han et al. [9] explore the magnitude-based pruning in conjunction with retraining, and report promising compression results without accuracy loss. It has also been validated that the sparse matrix-vector multiplication can further be accelerated by certain hardware design, making it more efficient than traditional CPU and GPU calculations [7]. The drawback of Han et al.'s method [9] is mostly its potential risk of irretrievable network damage and learning inefficiency.

Our research on network pruning is partly inspired by [9], not only because it can be very effective to compress DNNs, but also because it makes no assumption on the network structure. In particular, this branch of methods can be naturally combined with many other methods introduced above, to further reduce the network complexity. In fact, Han et al. [8] have already tested such combinations and obtained excellent results.

## 3   Dynamic Network Surgery

In this section, we highlight the intuition of our method and present its implementation details. In order to simplify the explanations, we only talk about the convolutional layers and the fully connected layers. However, as claimed in [8], our pruning method can also be applied to some other layer types as long as their underlying mathematical operations are inner products on vector spaces.

### 3.1   Notations

First of all, we clarify the notations in this paper. Suppose a DNN model can be represented as $\{\mathbf{W}_k : 0 \leq k \leq C\}$, in which $\mathbf{W}_k$ denotes a matrix of connection weights in the $k$th layer. For the fully connected layers with $p$-dimensional input and $q$-dimensional output, the size of $\mathbf{W}_k$ is simply $q_k \times p_k$. For the convolutional layers with learnable kernels, we unfold the coefficients of each kernel into a vector and concatenate all of them to $\mathbf{W}_k$ as a matrix.

In order to represent a sparse model with part of its connections pruned away, we use $\{\mathbf{W}_k, \mathbf{T}_k : 0 \leq k \leq C\}$. Each $\mathbf{T}_k$ is a binary matrix with its entries indicating the states of network connections, i.e., whether they are currently pruned or not. Therefore, these additional matrices can be considered as the mask matrices.

### 3.2   Pruning and Splicing

Since our goal is network pruning, the desired sparse model shall be learnt from its dense reference. Apparently, the key is to abandon unimportant parameters and keep the important ones. However, the parameter importance (i.e., the connection importance) in a certain network is extremely difficult

to measure because of the mutual influences and mutual activations among interconnected neurons. That is, a network connection may be redundant due to the existence of some others, but it will soon become crucial once the others are removed. Therefore, it should be more appropriate to conduct a learning process and continually maintain the network structure.

Taking the $k$th layer as an example, we propose to solve the following optimization problem:

$$\min_{\mathbf{W}_k, \mathbf{T}_k} L\left(\mathbf{W}_k \odot \mathbf{T}_k\right) \quad \text{s.t. } \mathbf{T}_k^{(i,j)} = \mathbf{h}_k(\mathbf{W}_k^{(i,j)}), \ \forall (i,j) \in \mathcal{I}, \tag{1}$$

in which $L(\cdot)$ is the network loss function, $\odot$ indicates the Hadamard product operator, set $\mathcal{I}$ consists of all the entry indices in matrix $\mathbf{W}_k$, and $\mathbf{h}_k(\cdot)$ is a discriminative function, which satisfies $\mathbf{h}_k(w) = 1$ if parameter $w$ seems to be crucial in the current layer, and 0 otherwise. Function $\mathbf{h}_k(\cdot)$ is designed on the base of some prior knowledge so that it can constrain the feasible region of $\mathbf{W}_k \odot \mathbf{T}_k$ and simplify the original NP-hard problem. For the sake of topic conciseness, we leave the discussions of function $\mathbf{h}_k(\cdot)$ in Section 3.3. Problem (1) can be solved by alternately updating $\mathbf{W}_k$ and $\mathbf{T}_k$ through the stochastic gradient descent (SGD) method, which will be introduced in the following paragraphs.

Since binary matrix $\mathbf{T}_k$ can be determined with the constraints in (1), we only need to investigate the update scheme of $\mathbf{W}_k$. Inspired by the method of Lagrange Multipliers and gradient descent, we give the following scheme for updating $\mathbf{W}_k$. That is,

$$\mathbf{W}_k^{(i,j)} \leftarrow \mathbf{W}_k^{(i,j)} - \beta \frac{\partial}{\partial(\mathbf{W}_k^{(i,j)} \mathbf{T}_k^{(i,j)})} L\left(\mathbf{W}_k \odot \mathbf{T}_k\right), \ \forall (i,j) \in \mathcal{I}, \tag{2}$$

in which $\beta$ indicates a positive learning rate. It is worth mentioning that we update not only the important parameters, but also the ones corresponding to zero entries of $\mathbf{T}_k$, which are considered unimportant and ineffective to decrease the network loss. This strategy is beneficial to improve the flexibility of our method because it enables the splicing of improperly pruned connections.

The partial derivatives in formula (2) can be calculated by the chain rule with a randomly chosen minibatch of samples. Once matrix $\mathbf{W}_k$ and $\mathbf{T}_k$ are updated, they shall be applied to re-calculate the whole network activations and loss function gradient. Repeat these steps iteratively, the sparse model will be able to produce excellent accuracy. The above procedure is summarized in Algorithm 1.

---

**Algorithm 1** Dynamic network surgery: the SGD method for solving optimization problem (1):

---

**Input: X**: training datum (with or without label), $\{\widehat{\mathbf{W}}_k : 0 \leq k \leq C\}$: the reference model, $\alpha$: base learning rate, $f$: learning policy.
**Output:** $\{\mathbf{W}_k, \mathbf{T}_k : 0 \leq k \leq C\}$: the updated parameter matrices and their binary masks.
Initialize $\mathbf{W}_k \leftarrow \widehat{\mathbf{W}}_k, \mathbf{T}_k \leftarrow \mathbf{1}, \forall 0 \leq k \leq C, \beta \leftarrow 1$ and iter $\leftarrow 0$
**repeat**
    Choose a minibatch of network input from **X**
    Forward propagation and loss calculation with $(\mathbf{W}_0 \odot \mathbf{T}_0),...,(\mathbf{W}_C \odot \mathbf{T}_C)$
    Backward propagation of the model output and generate $\nabla L$
    **for** $k = 0, ..., C$ **do**
        Update $\mathbf{T}_k$ by function $\mathbf{h}_k(\cdot)$ and the current $\mathbf{W}_k$, with a probability of $\sigma(\text{iter})$
        Update $\mathbf{W}_k$ by Formula (2) and the current loss function gradient $\nabla L$
    **end for**
    Update: iter $\leftarrow$ iter $+ 1$ and $\beta \leftarrow f(\alpha, \text{iter})$
**until** iter reaches its desired maximum

---

Note that, the dynamic property of our method is shown in two aspects. On one hand, pruning operations can be performed whenever the existing connections seem to become unimportant. Yet, on the other hand, the mistakenly pruned connections shall be re-established if they once appear to be important. The latter operation plays a dual role of network pruning, and thus it is called "network splicing" in this paper. Pruning and splicing constitute a circular procedure by constantly updating the connection weights and setting different entries in $\mathbf{T}_k$, which is analogical to the synthesis of excitatory and inhibitory neurotransmitter in human nervous system [17]. See Figure 2 for the overview of our method and the method pipeline can be found in Figure 1(a).

### 3.3 Parameter Importance

Since the measure of parameter importance influences the state of network connections, function $\mathbf{h}_k(\cdot), \forall 0 \leq k \leq C$, can be essential to our dynamic network surgery. We have tested several candidates and finally found the absolute value of the input to be the best choice, as claimed in [9]. That is, the parameters with relatively small magnitude are temporarily pruned, while the others with large magnitude are kept or spliced in each iteration of Algorithm 1. Obviously, the threshold values have a significant impact on the final compression rate. For a certain layer, a single threshold can be set based on the average absolute value and variance of its connection weights. However, to improve the robustness of our method, we use two thresholds $a_k$ and $b_k$ by importing a small margin $t$ and set $b_k$ as $a_k + t$ in Equation (3). For the parameters out of this range, we set their function outputs as the corresponding entries in $\mathbf{T}_k$, which means these parameters will neither be pruned nor spliced in the current iteration.

$$\mathbf{h}_k(\mathbf{W}_k^{(i,j)}) = \begin{cases} 0 & \text{if } a_k > |\mathbf{W}_k^{(i,j)}| \\ \mathbf{T}_k^{(i,j)} & \text{if } a_k \leq |\mathbf{W}_k^{(i,j)}| < b_k \\ 1 & \text{if } b_k \leq |\mathbf{W}_k^{(i,j)}| \end{cases} \tag{3}$$

### 3.4 Convergence Acceleration

Considering that Algorithm 1 is a bit more complicated than the standard backpropagation method, we shall take a few more steps to boost its convergence. First of all, we suggest slowing down the pruning and splicing frequencies, because these operations lead to network structure change. This can be done by triggering the update scheme of $\mathbf{T}_k$ stochastically, with a probability of $p = \sigma(\text{iter})$, rather than doing it constantly. Function $\sigma(\cdot)$ shall be monotonically non-increasing and satisfy $\sigma(0) = 1$. After a prolonged decrease, the probability $p$ may even be set to zero, i.e., no pruning or splicing will be conducted any longer.

Another possible reason for slow convergence is the vanishing gradient problem. Since a large percentage of connections are pruned away, the network structure should become much simpler and probably even much "thinner" by utilizing our method. Thus, the loss function derivatives are likely to be very small, especially when the reference model is very deep. We resolve this problem by pruning the convolutional layers and fully connected layers separately, in the dynamic way still, which is somehow similar to [9].

## 4 Experimental Results

In this section, we will experimentally analyse the proposed method and apply it on some popular network models. For fair comparison and easy reproduction, all the reference models are trained by the GPU implementation of Caffe package [12] with .prototxt files provided by the community.[2] Also, we follow the default experimental settings for SGD method, including the training batch size, base learning rate, learning policy and maximal number of training iterations. Once the reference models are obtained, we directly apply our method to reduce their model complexity. A brief summary of the compression results are shown in Table 1.

Table 1: Dynamic network surgery can remarkably reduce the model complexity of some popular networks, while the prediction error rate does not increase.

| model | Top-1 error | Parameters | Iterations | Compression |
|---|---|---|---|---|
| LeNet-5 reference | 0.91% | 431K | 10K | |
| LeNet-5 pruned | 0.91% | 4.0K | 16K | **108×** |
| LeNet-300-100 reference | 2.28% | 267K | 10K | |
| LeNet-300-100 pruned | 1.99% | 4.8K | 25K | **56×** |
| AlexNet reference | 43.42% | 61M | 450K | |
| AlexNet pruned | 43.09% | 3.45M | 700K | **17.7×** |

## 4.1 The Exclusive-OR Problem

To begin with, we consider an experiment on the synthetic data to preliminary testify the effectiveness of our method and visualize its compression quality. The exclusive-OR (XOR) problem can be a good option. It is a nonlinear classification problem as illustrated in Figure 3(a). In this experiment, we turn the original problem to a more complicated one as Figure 3(b), in which some Gaussian noises are mixed up with the original data $(0,0), (0,1), (1,0)$ and $(1,1)$.

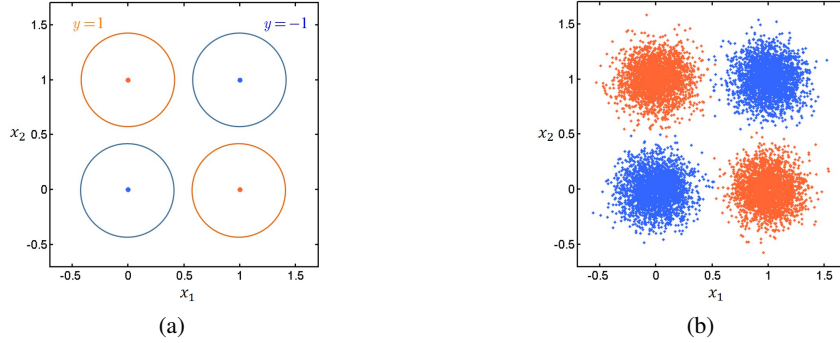

(a)  (b)

Figure 3: The Exclusive-OR (XOR) classification problem (a) without noise and (b) with noise.

In order to classify these samples, we design a network model as illustrated in the left part of Figure 4(a), which consists of 21 connections and each of them has a weight to be learned. The sigmoid function is chosen as the activation function for all the hidden and output neurons. Twenty thousand samples were randomly generated for the experiment, in which half of them were used as training samples and the rest as test samples.

By 100,000 iterations of learning, this three-layer neural network achieves a prediction error rate of 0.31%. The weight matrix of network connections between input and hidden neurons can be found in Figure 4(b). Apparently, its first and last row share the similar elements, which means there are two hidden neurons functioning similarly. Hence, it is appropriate to use this model as a compression reference, even though it is not very large. After 150,000 iterations, the reference model will be compressed into the right side of Figure 4(a), and the new connection weights and their masks are shown in Figure 4(b). The grey and green patches in $\mathbf{T}_1$ stand for those entries equal to one, and the corresponding connections shall be kept. In particular, the green ones indicate the connections were mistakenly pruned in the beginning but spliced during the surgery. The other patches (i.e., the black ones) indicate the corresponding connections are permanently pruned in the end.

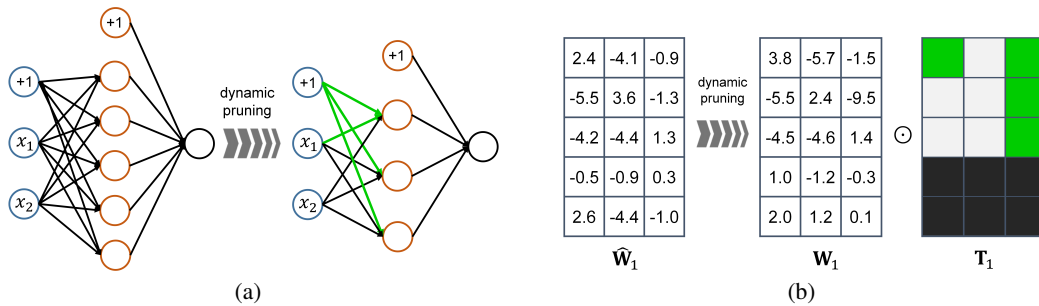

(a)  (b)

Figure 4: Dynamic network surgery on a three-layer neural network for the XOR problem. (a): The network complexity is reduced to be optimal. (b) The connection weights are updated with masks.

The compressed model has a prediction error rate of 0.30%, which is slightly better than that of the reference model, even though 40% of its parameters are set to be zero. Note that, the remaining hidden neurons (excluding the bias unit) act as three different logic gates and altogether make up

the XOR classifier. However, if the pruning operations are conducted only on the initial parameter magnitude (as in [9]), then probably four hidden neurons will be finally kept, which is obviously not the optimal compression result.

In addition, if we reduce the impact of Gaussian noises and enlarge the margin between positive and negative samples, then the current model can be further compressed, so that one more hidden neuron will be pruned by our method.

So far, we have carefully explained the mechanism behind our method and preliminarily testified its effectiveness. In the following subsections, we will further test our method on three popular NN models and make quantitative comparisons with other network compression methods.

## 4.2 The MNIST database

MNIST is a database of handwritten digits and it is widely used to experimentally evaluate machine learning methods. Same with [9], we test our method on two network models: LeNet-5 and LeNet-300-100.

LeNet-5 is a conventional CNN model which consists of 4 learnable layers, including 2 convolutional layers and 2 fully connected layers. It is designed by LeCun et al. [15] for document recognition. With 431K parameters to be learned, we train this model for 10,000 iterations and obtain a prediction error rate of 0.91%. LeNet-300-100, as described in [15], is a classical feedforward neural network with three fully connected layers and 267K learnable parameters. It is also trained for 10,000 iterations, following the same learning policy as with LeNet-5. The well trained LeNet-300-100 model achieves an error rate of 2.28%.

With the proposed method, we are able to compress these two models. The same batch size, learning rate and learning policy are set as with the reference training processes, except for the maximal number of iterations, which is properly increased. The results are shown in Table 1. After convergence, the network parameters of LeNet-5 and LeNet-300-100 are reduced by a factor of $108\times$ and $56\times$, respectively, which means less than 1% and 2% of the network connections are kept, while the prediction accuracies are as good or slightly better.

Table 2: Compare our compression results on LeNet-5 and LeNet-300-100 with that of [9]. The percentage of remaining parameters after applying Han et al's method [9] and our method are shown in the last two columns.

| Model | Layer | Params. | Params.% [9] | Params.% (Ours) |
|---|---|---|---|---|
| LeNet-5 | conv1 | 0.5K | $\sim 66\%$ | 14.2% |
| | conv2 | 25K | $\sim 12\%$ | 3.1% |
| | fc1 | 400K | $\sim 8\%$ | 0.7% |
| | fc2 | 5K | $\sim 19\%$ | 4.3% |
| | Total | 431K | $\sim 8\%$ | **0.9%** |
| LeNet-300-100 | fc1 | 236K | $\sim 8\%$ | 1.8% |
| | fc2 | 30K | $\sim 9\%$ | 1.8% |
| | fc3 | 1K | $\sim 26\%$ | 5.5% |
| | Total | 267K | $\sim 8\%$ | **1.8%** |

To better demonstrate the advantage of our method, we make layer-by-layer comparisons between our compression results and Han et al.'s [9] in Table 2. To the best of our knowledge, their method is so far the most effective pruning method, if the learning inefficiency is not a concern. However, our method still achieves at least 4 times the compression improvement against their method. Besides, due to the significant advantage over Han et al.'s models [9], our compressed models will also be undoubtedly much faster than theirs.

## 4.3 ImageNet and AlexNet

In the final experiment, we apply our method to AlexNet [13], which wins the ILSVRC 2012 classification competition. As with the previous experiments, we train the reference model first.

Without any data augmentation, we obtain a reference model with 61M well-learned parameters after 450K iterations of training (i.e., roughly 90 epochs). Then we perform the network surgery on it. AlexNet consists of 8 learnable layers, which is considered to be deep. So we prune the convolutional layers and fully connected layers separately, as previously discussed in Section 3.4. The training batch size, base learning rate and learning policy still keep the same with reference training process. We run 320K iterations for the convolutional layers and 380K iterations for the fully connected layers, which means 700K iterations in total (i.e., roughly 140 epochs). In the test phase, we use just the center crop and test our compressed model on the validation set.

Table 3: The comparison of different compressed models, with Top-1 and Top-5 prediction error rate, the number of training epochs and the final compression rate shown in the table.

| Model | Top-1 error | Top-5 error | Epochs | Compression |
|---|---|---|---|---|
| Fastfood 32 (AD) [21] | 41.93% | - | - | $2\times$ |
| Fastfood 16 (AD) [21] | 42.90% | - | - | $3.7\times$ |
| Naive Cut [9] | 57.18% | 23.23% | 0 | $4.4\times$ |
| Han et al. [9] | 42.77% | 19.67% | $\geq 960$ | $9\times$ |
| Dynamic network surgery (Ours) | 43.09% | 19.99% | $\sim 140$ | $\mathbf{17.7\times}$ |

Table 3 compares the result of our method with some others. The four compared models are built by applying Han et al.'s method [9] and the adaptive fastfood transform method [21]. When compared with these "lossless" methods, our method achieves the best result in terms of the compression rate. Besides, after acceptable number of epochs, the prediction error rate of our model is comparable or even better than those models compressed from better references.

In order to make more detailed comparisons, we compare the percentage of remaining parameters in our compressed model with that of Han et al.'s [9], since they achieve the second best compression rate. As shown in Table 4, our method compresses more parameters on almost every single layer in AlexNet, which means both the storage requirement and the number of FLOPs are better reduced when compared with [9]. Besides, our learning process is also much more efficient thus considerable less epochs are needed (as least 6.8 times decrease).

Table 4: Compare our method with [9] on AlexNet.

| Layer | Params. | Params.% [9] | Params.% (Ours) |
|---|---|---|---|
| conv1 | 35K | $\sim 84\%$ | 53.8% |
| conv2 | 307K | $\sim 38\%$ | 40.6% |
| conv3 | 885K | $\sim 35\%$ | 29.0% |
| conv4 | 664K | $\sim 37\%$ | 32.3% |
| conv5 | 443K | $\sim 37\%$ | 32.5% |
| fc1 | 38M | $\sim 9\%$ | 3.7% |
| fc2 | 17M | $\sim 9\%$ | 6.6% |
| fc3 | 4M | $\sim 25\%$ | 4.6% |
| Total | 61M | $\sim 11\%$ | **5.7%** |

## 5  Conclusions

In this paper, we have investigated the way of compressing DNNs and proposed a novel method called dynamic network surgery. Unlike the previous methods which conduct pruning and retraining alternately, our method incorporates connection splicing into the surgery and implements the whole process in a dynamic way. By utilizing our method, most parameters in the DNN models can be deleted, while the prediction accuracy does not decrease. The experimental results show that our method compresses the number of parameters in LeNet-5 and AlexNet by a factor of $\mathbf{108\times}$ and $\mathbf{17.7\times}$, respectively, which is superior to the recent pruning method by considerable margins. Besides, the learning efficiency of our method is also better thus less epochs are needed.

## Footnotes

[2]Except for the simulation experiment and LeNet-300-100 experiments which we create the .prototxt files by ourselves, because they are not available in the Caffe model zoo.

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
