[Reviews · NeurIPS 2016]

Reviewer 1

Summary

The paper advances the state of the art in network compression. This is critical for the purpose of reducing memory and computation (in testing) overhead for mobile devices, considering the huge size of the deep networks considered in the literature. The main idea is adding to the usual pruning methods the option to recover an existing connection if it is useful at any time. The main method is to consider a mask for the weight matrix that can then be used in the optimization of the new objective function which is solved by the method of Lagrange multipliers. The choice of the mask matrix is given by a boolean function parameterized by two thresholds which are layer dependent and are set based on the mean and variance of the connection weights. The experiments show extremely good performance in terms of compression ratio and convergence speed, without any loss of accuracy.

Qualitative Assessment

The approach adds a complementary process to the pruning process. The results are very important as many researchers can benefit from an implementation of this technique, e.g. mobile user making use of deep nets. The paper addresses a difficult problem in a better way and it advances the state of the art in demonstrable ways. It provides a unique conclusion and unique practical approach. The part describing the important parameters a_k and b_k is not clear, and does not show how exactly you get/select these values, which makes the paper not reproducible. This is why the score of 2 for Clarity and Technical. I think you should be more specific when saying a Boolean-valued function has gradient zero (if exists). Because the function is piecewise constant, it has a number of discrete points where it is not differentiable. Language corrections: Line 23: 'What's more' should be 'Moreover'. Line 39: 'limit of pruning method' should be 'limit of the pruning method' Line 42: 'conducted on two' should be 'conducted with two' Line 44: 'responsible to' should be 'responsible for' Line 52: 'system' should be 'systems' Line 87: 'assumption to' should be 'assumption on'. Line 137: 'shows' should be 'is shown' Line 237: 'Then it is the network surgery' is ambiguous. Should reformulate the phrase starting at line 240. Line 259: Definitely not 'most'. Maybe a lot, a significant amount, but not most. Missing notations (they are defined after they are used): Line 98: What is C ? Algorithm 1: What is \sigma(iter)?

Confidence in this Review

2-Confident (read it all; understood it all reasonably well)


Reviewer 2

Summary

This paper studies how to reduce neural network size, by adding a learnable binary mask to each weight parameter. If the binary mask is 0, then the weight is pruned.

Qualitative Assessment

To reduce network size without sacrificing accuracy, this paper proposes a simple but effective idea: adding a binary mask to each weight parameter and learning the binary mask automatically from data. Empirical evaluation on several datasets shows the promising effectiveness of the proposed method. My major concern is that the experiments are not sufficient convincing. Many recent works have been devoted to this paper (reducing network size), but they are not included for comparison. Here are some relevant papers. [1] Deep Compression: Compressing Deep Neural Networks with Pruning, Trained Quantization and Huffman Coding, 2016 [2] Training Convolutional Neural Networks with Low-rank Filters for Efficient Image Classification, 2016 [3] Convolutional neural networks with low-rank regularization, 2016 [4] Diversity Networks, 2016 Given the lacking of comparisons with more recent and competitive baseline methods, I vote for rejection.

Confidence in this Review

2-Confident (read it all; understood it all reasonably well)


Reviewer 3

Summary

This work deals with the compression of neural networks by means of network "surgery". The authors describe two operations performed on the network: pruning and slicing. The pruning. These operations are related to one another through a binary masking matrix which is used on the forward pass during training. In particular, each weight matrix is masked based on the absolute values of its entries, causing a large fraction of the parameters to be either zero or one when computing the loss function. During the backward pass the gradient updates are kept in the continuous (unmasked) parameter space.

Qualitative Assessment

The strength of this work lies primarily in the experimental results which show state-of-the-art compression gains for neural networks. Although there is not much in the way of theory to supplement these gains, the algorithms presented in this paper have a high chance of being adopted by the field at large. Indeed, there already appears to be a common trend in the compression community, which this work is capitalizing upon. It seems, that quantizing (or in this case sparsifying) the forward pass for inference during training, while keeping the gradients in the original parameter space [1,2,3]. A deeper connection might be made to this work in the text. At the same time the lagrangian in equation (2) seems somewhat superfluous, as it complicates the description of an otherwise elegant algorithm to achieve impressive compression gains. One weakness of this paper is that it does not present results in terms of CPU runtime. The low-rank approaches to model compression have the advantage that ultimately using the model at inference time requires simple matrix multiplications. The authors leave open the question of whether the sparsity they have gained with their approach yields speed improvements in addition to the clear wins with respect to memory. [1] "BinaryConnect: Training Deep Neural Networks with binary weights during propagations", Matthieu Courbariaux, Yoshua Bengio, Jean-Pierre David [2] "Fixed-point feedforward deep neural network design using weights -1, 0, and +1", K. Hwang and W. Sung [3] “Bitwise neural networks,” M. Kim and P. Smaragdis.

Confidence in this Review

2-Confident (read it all; understood it all reasonably well)


Reviewer 4

Summary

This paper presents a new iterative algorithm for pruning deep networks, which is experimentally verified to be able to provide strong compression rate.

Qualitative Assessment

The authors present an overall decent empirical improvement in network pruning compared to existing methods. The paper may be further improved if experimental results on even larger networks (e.g. VGG networks) can be added. My main complaint however is that the authors didn't provide any theoretical analysis to support Algorithm 1's convergence property and why it can prevent the "irretrievable damage and learning inefficiency" scenarios, since it's unclear whether Algorithm 1 can actively avoid being trapped in such "local minima". Also, Eq. (1) and (4) together form a confusing loop.

Confidence in this Review

2-Confident (read it all; understood it all reasonably well)


Reviewer 5

Summary

The paper presents a method to prune neural network weights with an aim to reduce the number of parameters. A training methodology superior to that of Han et al is presented, and as a result they obtain much better compression results.

Qualitative Assessment

The results of this method are indeed quite impressive. It obtains considerably better compression performance when compared to Han et al., the primary paper which is being compared. However, the formulation seems quite problematic and unclear. 1) Eqn (1) has to be optimized w.r.t two terms - W and T. However, given that there is a strict equality constraint between these two terms, optimizing one would automatically determine the other. 2) the Lagrangian obtained in Eqn (2) seems incorrect. The regularizer would instead look like (h(W) - T) without the squared term. 3) Eqn (1) is considered to be the objective function to solve. However, it is not clear how the algorithm minimizes Eqn (1). In addition, a regularization constant \gamma is defined in eqn(2), but never used in the paper. Moreover, this method inherits a major shortcoming of Han et al, which is that it requires setting thresholds for each layer. I presume that these thresholds were chosen manually each time to get optimal compression. Further, there is also \sigma (iter) - the probability of updating T. Line 163: Authors mention vanishing gradient problem due to thinner nets. However, because of fewer parameters, shouldn’t each parameter have a larger gradient, hence making it easier to train? Even if this is not true, it is not clear how training convolutional and fully connected (FC) layers separately solves the problem, since the gradients for the conv layers have to travel through the FC layers in either case. Please clarify. Overall, it is an extension of an existing method to enhance compression results. However, some technical issues and the heuristic nature of the method remain causes of concern.

Confidence in this Review

2-Confident (read it all; understood it all reasonably well)


Reviewer 6

Summary

This paper proposes a method for DNN compression which is essential for reducing the storage and processing power requirements of very deep networks. The state-of-the-art method on deep network compression uses a variety of hashing and quantization mechanisms followed by a network pruning and retraining phase. In this paper, the authors propose a method for network pruning and retraining phase of the deep network compression method of Han et al. by introducing the option of restoring previously pruned connections. They show the efficiency of their method through experimental evaluation on several benchmark datasets. Reported results suggest that their method performs better in terms of compression rate vs. accuracy trade-off in most of the benchmarks.

Qualitative Assessment

I think having the option of restoring the pruned connections is a very good idea and apparently it works well on the experimented benchmarks. On the other hand, the argument that Han et al's method can cause permanent damage in the network does not seem exactly accurate. Because, actually their network performs better than yours in LeNet-300-100 Ref vs LeNet-300-100 Pruned (with 0.05 loss whereas your network has 0.29 loss). This is also the case for AlexNet. Their accuracy gap is better than yours. So, I don't think there is a permanent damage issue there. Secondly, they make multiple sessions of pruning and retraining so the network always has the opportunity to compensate for the erroneous pruning. Your method is better in terms of run time because it requires less iterations, yet, it also brings additional space requirements that do not exist in Han et al.'s method. I think a small discussion on that issue would be great. You report results on all the benchmark datasets Han et al. used except for VGG-16. I wonder what is the reason for skipping VGG-16, which is the deepest and largest network. In Line 100, you state "For the convolutional layers with learnable kernels, we unfold the coefficients of each kernel into a vector and concatenate all of them to Wk as a matrix." What is the reason for this? I think the manuscript lacks bunch of details about the parameter selection which might be very relevant to the audience. In Algorithm 1, the parameter \alpha is used for updating the learning \beta. This parameter is not explained. Which learning rate update scheme do you use? For updating the parameters, you state that you have tested several candidates and decided upon magnitude-based elimination. Can you elaborate on what other schemes you have tested? On line 146, you say "the absolute value of the input". I think it should be the absolute value of the weight or parameter. The input is not the same as weights. How you decide on thresholds for parameter importance is left unclear. Can you provide more details on how you select a_k, b_k and t? Also, what scheme do you use to decay the update probability \sigma(iter) On line 188, you say three-layer network but according to your definition this is a two layer network. In Han et al. paper, they change the dropout probability during compression. How do you deal with dropout in your method? In Section 4.1, there is a toy example that depicts how your method works. It'd be great to see how Han et al.'s method would do on this example. Minor comments: The figures look really nice in the paper but they are unreadable when printed BW.

Confidence in this Review

2-Confident (read it all; understood it all reasonably well)